# Effects of the In Ovo Injection of Vitamin D_3_ and 25-Hydroxyvitamin D_3_ in Ross 708 Broilers Subsequently Challenged with Coccidiosis: II Immunological and Inflammatory Responses and Small Intestine Histomorphology [note 1]

**DOI:** 10.3390/ani12081027

**Published:** 2022-04-14

**Authors:** Seyed Abolghasem Fatemi, Katie E. C. Elliott, Ken S. Macklin, Abiodun Bello, Edgar David Peebles

**Affiliations:** 1Department of Poultry Science, Mississippi State University, Mississippi State, MS 39762, USA; katie@poultry.msstate.edu (K.E.C.E.); d.peebles@msstate.edu (E.D.P.); 2Poultry Research Unit, USDA-ARS, Mississippi State, MS 39762, USA; 3Department of Poultry Science, College of Agriculture, Auburn University, Auburn, AL 36849, USA; macklks@auburn.edu; 4Department of Agricultural, Food and Nutritional Science, University of Alberta, Edmonton, AB T6G 2P5, Canada; abiodun.bello@iff.com

**Keywords:** in ovo injection, 25-hydroxyvitamin D_3_, small intestine morphology, inflammatory response, and coccidiosis

## Abstract

**Simple Summary:**

Vitamin D_3_ sources serve as immunomodulators and improve intestinal morphology, which can promote broiler performance. The in ovo injection of vitamin D_3_ sources has been shown to enhance immunity as well as histomorphological variables in unchallenged conditions. One of the main diseases affecting poultry production is coccidiosis. Because of this, the current study was designed to determine the effects vitamin D_3_ (D_3_) and 25-hydroxyvitamin D_3_ (25OHD_3_) alone or together on the inflammatory reaction and small intestine morphology of broilers that were challenged with coccidiosis. In this study, it is shown that the in ovo administration of 2.4 μg of 25OHD_3_ alone increased villus length to crypt depth ratio (VCR) with the D_3_ + 25OHD_3_ treatment being intermediate two weeks post-challenge (28 day of age). Furthermore, chickens that received of 25OHD_3_ alone experienced lower plasma nitric oxide concentration as a systemic inflammatory indicator in comparison to all other treatments. It is concluded that the in ovo injection of 2.4 μg of 25OHD_3_ at 18 days of incubation can enhance intestinal histomorphology as well as inflammatory reaction of broilers when infected with coccidiosis.

**Abstract:**

In broilers challenged with coccidiosis, effects of in ovo vitamin D_3_ (D_3_) and 25-hydroxyvitamin D_3_ (25OHD_3_) administration on their inflammatory response and small intestine morphology were evaluated. At 18 d of incubation (doi), a 50 μL volume of the following 5 in ovo injection treatments was administrated: non-injected (1) and diluent injected (2) controls, or diluent injection containing 2.4 μg D_3_ (3) or 2.4 μg 25OHD_3_ (4), or their combination (5). Four male broilers were randomly allocated to each of eight isolated replicate wire-floored battery cages at hatch, and birds were challenged at 14 d of age (doa) with a 20x live coccidial vaccine dosage. One bird from each treatment–replicate (40 birds in each of 8 replicates per treatment) was bled at 14 and 28 doa in order to collect blood for the determination of plasma IL-1β and nitric oxide (NO) concentrations. The duodenum, jejunum, and ilium from those same birds were excised for measurement of villus length, crypt depth, villus length to crypt depth ratio (VCR), and villus surface area. In ovo injection of 2.4 μg of 25OHD_3_ resulted in a reduction in plasma NO levels as compared to all other treatments at 28 doa. Additionally, duodenal VCR increased in response to the in ovo injection of 25OHD_3_ when compared to the diluent, D_3_ alone, and the D_3_ + 25OHD_3_ combination treatments at two weeks post-challenge (28 doa). Therefore, it can be concluded that 2.4 μg of 25OHD_3_, when administrated in ovo at 18 doi, may be used to decrease the inflammatory reaction as well as to enhance the small intestine morphology of broilers during a coccidiosis challenge.

## 1. Introduction

Coccidiosis is a parasitic infection that still is considered one of the main diseases affecting the performance of poultry reared under intensive production systems. Györke et al. [1] estimated that the total economic losses due to coccidiosis are approximately EUR 3162.4 (USD 3771.64) per flock and that coccidiosis overall led to a 34.8% increase in mortality and a 65.2% depression in feed efficiency. Coccidiosis has been shown to increase the feed conversation ratio (FCR) and reduce the feed intake and body weight (BW) gain of broilers due to deteriorative effects on their immunity and small intestine histomorphology [2,3]. Birds diagnosed with coccidiosis exhibit massive heterophil and mononuclear cell infiltration as well as severe damage to their intestinal villi. Morphological degeneration of the small intestine has been reported due to pathomorphological alterations caused by coccidiosis. More specifically, it has been shown that the villus height to crypt depth ratio decreases during a coccidiosis infection [4]. An increased villus height is associated with increased nutrient absorption [5], and a longer crypt depth is due to increased enterocyte turnover in the intestine, leading to increased gut energy requirements [6]. 

Vitamin D_3_ (D_3_) and its metabolite, 25-hydroxyvitamin D_3_ (25OHD_3_), have been shown to affect small intestine morphology and the humoral and inflammatory responses of broiler chickens [7,8,9]. Vitamins D_3_ and 25OHD_3_ are multifunctional nutrients that are involved in calcium and phosphorus absorption [10], bone metabolism and mineralization [11], muscle development [12], immunity [7,8,9,13] and small intestine morphology [7,14]. As compared to D_3_, 25OHD_3_ has been shown to be more effective in promoting the performance and immunity of broiler chickens during a coccidiosis infection [8,13]. The BW gain of infected birds is increased when they are fed 25OHD_3_ rather than D_3._ Additionally, the pro-inflammatory responses decrease in infected birds fed 25OHD_3_ in comparison to those fed D_3_ at the same levels of inclusion.

In ovo injection is a novel technology which has been widely used in the broiler industry to promote the immunity and growth of broiler embryos [15]. In addition, in ovo injection of several nutrients has been shown to improve the immunity [16], antioxidant capacity [17], and small intestine morphology [18] of broilers. Moreover, the in ovo injection of 2.4 μg of 25OHD_3_ has previously exhibited promising results concerning the hatchling quality [19], live performance [20,21,22,23], breast meat yield [21,22,23], small intestine morphology [18], and immunity [18,21] of Ross 708 broilers reared under normal conditions. Effects of the in ovo injection of a coccidiosis vaccine or a vitamin D source on broiler performance have also been previously reported. The in ovo administration of a coccidiosis vaccine has been shown to increase the oocyst shedding [24] and decrease the early posthatch [25] and late posthatch [26] performances of broilers. The in ovo injection of 25OHD_3_ has been observed to increase the bone quality of male broilers [27]. Additionally, preliminary work in our laboratory suggests that the humoral immunity and small intestine morphology of broilers are improved in response to the administration of 2.4 μg of 25OHD_3_ in the amnion at 18 d of incubation (doi). Impacts of in ovo injection of any nutrient or vaccine on small intestine morphology or immunity in chickens during a field challenge of coccidiosis have not been investigated. Therefore, the objective of this study was to determine the effects of in ovo administration of two vitamin D sources (D_3_ and 25OHD_3_) on the small intestine histomorphology and inflammatory reaction of broilers challenged by coccidiosis. 

## 2. Material and Methods

### 2.1. Experimental and Incubation Conditions

From Ross 708 broiler breeder hens that were 35 weeks of age and that were housed in a commercial facility, fertile broiler hatching eggs were obtained. Under standard conditions (37.5 °C and 29.4 °C dry and wet bulb temperatures, respectively) in a Chick Master Incubator (Chick Master Incubator Company, Medina, OH, USA), a total of 1000 eggs were set and incubated. Fifty eggs were assigned to each of 5 preassigned treatment groups that were randomly distributed among 20 incubator tray levels. In ovo injection treatments were prepared in accordance with the procedure described by Fatemi et al. [19,20] and a 50 μL solution volume was administrated targeting the amnion of each egg at 18 doi using a Zoetis Inovoject m (Zoetis, Parsippany, NJ, USA) multi-egg injection machine. The injection treatments that were delivered to the broiler hatching eggs containing live embryos at 18 d of incubation (doi) were non-injected (1) and diluent injected (2) controls, or injected with diluent containing 2.4 μg D_3_ (3) or 2.4 μg 25OHD_3_ (4), or a combination thereof (5). At hatch (21 doi), chicks from each replicate basket were pooled within their respective treatment group, and for each of 5 in ovo injection treatments, 4 male broilers were randomly allocated to each of 8 isolated replicate wire-floored battery cages (0.76 m × 0.46 m (0.35 m^2^)) per treatment in each of 2 separate isolated rooms. In each room, the cages containing birds in the non-injected treatment group were separated from the battery cages containing birds in the in ovo-injected treatments to eliminate their exposure to coccidial oocysts. The in-ovo-injected treatments were randomly assigned to the remaining battery cages. The environmental temperature and lighting in the rooms were maintained as specified by the primary breeder company [28] that met or exceeded NRC recommendation [29]. Chicks had ad libitum access to feed and fresh water. In order to meet Ross 708 commercial guidelines [23,28], a Mississippi State University basal corn–soybean diet formulation was used in the form of crumbles from 0 to 14 d of age (doa) and as pellets from 15 to 41 doa.

### 2.2. Coccidiosis Challenge

A coccidial challenge infection was induced using 20,000 live *Eimeria* (*E*) species, including *acervulina, maxima, mivati,* and *tenella* (Coccivac-B52, Intervet Inc. Omaha, NE, USA) suspended in 1 mL of distilled water. Challenges were administrated to all broiler chicks housed in battery cages in the diluent, D_3_, 25OHD_3_, and D_3_ + 25OHD_3_ in ovo injection treatment groups at 14 doa. The chicks in the uninjected treatment group were left unchallenged. At 14 and 28 doa, one bird from each treatment-replicate battery cage were randomly selected and individually weighed prior to subsequent immunological and small intestine morphological testing. 

### 2.3. Organ Characteristics, and Immunological and Inflammatory Response Assessments

In each treatment-replicate cage (40 total birds) in each room, one bird was randomly selected for sampling at 14 and 28 doa. After being weighed and euthanized, the lengths and weights of the duodenum, jejunum, ileum, and cecum and weights of the bursa, spleen, liver, and entire small intestine of each bird were obtained. Organ weights were calculated as percentages of live BW. Furthermore, blood from the brachial vein of each bird sampled was collected and plasma was then subsequently extracted for immunological assay. Plasma IL-1β concentration was measured according to the manufacturer’s protocol ((Chicken interleukin 1 beta (IL-1β) ELISA kit Cat. NO. MBS261585); MyBiossource, San Diego, CA, USA). The IL-1β ELISA assay was performed as introduced by Fatemi et al. [18]. Changes to that procedure included the use of biotinylated chicken IL-1β antibody and specific IL-1β monoclonal antibody precoated plates. Plasma nitric oxide (NO) concentration was determined according to the procedure described by Bowen et al. [30]. Optical density (OD) at 450 nm (OD450) for IL-1β and NO was measured with a SpectraMax M5 Microplate Reader (Molecular Devices, San Jose, CA, USA). 

### 2.4. Small Intestine Morphology

From the same birds sampled for immunological and inflammatory assessment, intestine samples were collected before challenge (14 doa) and 2 weeks post-challenge (28 doa). Tissue samples (2 cm in length) were obtained from the midpoint of the duodenum, jejunum, and ileum and then fixed in 10% formalin for subsequent morphological examination. Intestinal samples were prepared for histomorphological analysis according to the procedure described by Wang et al. [4]. Villus length (VL), crypt depth (CD), villus length to crypt depth ratio (VCR), and villus surface area (VSA) were quantified in the duodenum, jejunum, and ileum as described by Nain et al. (2012) and Fatemi et al. [18]. Microscopic images were obtained using a light microscope at a 40× magnification (Micromaster, Fisher Scientific, Pittsburgh, PA, USA), and ImageJ software (Wayne Rasband, NIMH, Bethesda, MD, USA) was used to measure the length of the villus components.

All intestinal sections from each bird were contained on an individual slide. The average of the measurements at 3 different locations of each intestinal section was used for statistical analysis. Villus height was measured from the tip to the base of the lamina propria. Villus width (VW) was also calculated as the average of the weights of the top third and the bottom third of the villus, and then CD was considered as the length between the base of the villus to the base of the mucosa. The ratio of VL to CD was calculated by dividing villus height by CD. Villus surface area (VSA) was calculated using the following formula described by Nain et al. [31].
Villus surface area (VSA) = 2π × (average VW/2) × VL;(1)
where average VW is the average of 6 measurements per bird (2 width measurements from the three sample sections examined on each slide).

### 2.5. Statistical Analysis

The experimental unit was the wire-floored battery cage for the blood and small intestine data. The experimental design was a completely randomized block. Room was the blocking factor with all treatments randomly represented in each of the 2 blocks. Because there was only one replicate unit in the non-injected treatment group, that treatment was not included in the statistical analysis of the effects of in ovo injection treatment after a coccidiosis challenge. All data were analyzed with a one-way ANOVA using the procedure for linear mixed models (PROC GLIMMI) of SAS 9.4© [32]. Treatment differences were deemed significant at *p* < 0.05. The following model was used for analysis of the data:Y_ij_ = μ + B_i_ + T_j_ + E_ij_,(2)
where μ was the population mean; B_i_ was the block factor (i = 1 to 2); Tj was the effect of in ovo injection treatment (j = 1 to 5); E_ij_ was the residual error.

## 3. Results

### 3.1. Organ Characteristics

No significant treatment differences were observed for the BW; the lengths and relative weights of the duodenum, jejunum, and ileum; and relative weights of the spleen, liver, and cecum at 14 and 28 doa. However, ceca length at 14 doa was longest for the D_3_ + 25OHD_3_-injected birds and it was shortest for the non-injected (unchallenged) birds with consecutive and significant increases in the diluent, D_3_, and 25OHD_3_ treatments (Table 1). Additonally, at 28 doa, birds that were injected with 25OHD_3_ had longer ceca as compared to those that were diluent-injected, with the D_3_ + 25OHD_3_ treatment being intermediate. Additionally, there was a trend toward a treatment effect on spleen weight relative to BW that approached significance (*p* = 0.055). Birds that received 25OHD_3_ alone also tended to have higher spleen weights as compared to the non-injected and D_3_ + 25OHD_3_-injected birds (Table 1). At 28 doa or 2 weeks post-challenge, birds injected with 25OHD_3_ had lower bursa weights relative to BW as compared to all other in ovo treatments (Table 1).

### 3.2. Immunological and Inflammatory Responses

No significant treatment effects were observed for the immunological measurements at 14 doa (before coccidiosis challenge) (Table 2). However, there was a significant treatment difference for plasma NO concentration (*p* = 0.028) at 28 doa (after coccidiosis challenge). The in ovo injection of 25OHD_3_ alone reduced plasma NO levels as compared to all other in ovo treatments (Table 2).

### 3.3. Small Intestine Morphology

At 14 doa, the in ovo injection of 25OHD_3_ alone resulted in higher duodenal and ileal VCR as compared to all other treatments, and non-injected birds had a lower ileal VCR in comparison to birds that received in ovo injections of 25OHD_3_ and D_3_ + 25OHD_3_. A greater jejunal VCR belonged to 25OHD_3_ in ovo-injected birds alone as compared to those that were non-injected or that were in ovo-injected with D_3_ alone (Table 3). There tended to be treatment differences for duodenal (*p* = 0.076) and jejunal (*p* = 0.056) CD that approached significance. The in ovo injection of 25OHD_3_ alone tended to decrease duodenal CD in comparison to the diluent-injected control group, and it tended to reduce jejunal CD in comparison to the D_3_ alone treatment (Table 3). At 28 doa, VL and VCR in the duodenum were increased by 25OHD_3_ injection in comparison to all other injection treatment groups. Moreover, in ovo injection of 25OHD_3_ alone increased ileal VCR in comparison to all other treatments except for the D_3_ + 25OHD_3_ treatment. The ileal VCR of the D_3_-alone injected treatment group was also significantly lower than that of the D_3_ + 25OHD_3_ injection treatment (Table 4). Furthermore, there tended to be treatment effects on VCR (*p* = 0.058) in the jejunum, and VL (*p* = 0.056) and CD (*p* = 0.054) in the ileum. The in ovo injection of 25OHD_3_ alone tended to increase jejunal VCR and the in ovo injection of D_3_ alone tended to increase ileal CD in comparison to all other in ovo treatments. Also, birds in ovo-injected with 25OHD_3_ alone tended to have a greater ileal VL when compared to all other treatment groups (Table 4).

## 4. Discussion

The aim of this study was to investigate the impact of the in ovo injection of two sources of vitamin D (D_3_ and 25OHD_3_) on the small intestine morphology and inflammatory response of broilers during a coccidiosis challenge. Coccidiosis in poultry is economically the most important disease affecting the performance of poultry reared under intensive production systems. In poultry, most coccidiosis species belong to the genus *Eimeria* and damage various sections in the intestine. The infection process is quick (4 to 7 d) and is identified by monitoring parasite recycling in host cells with significant damage to the intestinal mucosa. Poultry coccidia are mainly host-specific, and the various species are localized in specific parts of the intestine. Small intestine histomorphology is also associated with broiler performance during an *Eimeria* infection. Negative effects on small intestine morphology and broiler performance have been observed in coccidiosis-infected broilers in comparison to those that were non-infected [4]. In the current study, impairments in small intestine morphology were observed in coccidiosis-infected birds. This would provide a partial reason for the decline in broiler performance in coccidiosis-infected birds that was observed by Fatemi et al. [22]. However, in comparison to the injection of diluent or D_3_ alone in that study, the BW and BW gain of challenged broilers were increased in response to the in ovo injection of 2.4 μg of 25OHD_3_. The improvement in the live performance of broilers could be partially linked to an enhancement in small intestine morphology.

In poultry, coccidiosis has been shown to increase villus atrophy, crypt hyperplasia [33], and leukocyte infiltration [34]. In addition, coccidiosis has been shown to impair small intestine morphology by decreasing VL [35] and VCR [33,35] and increasing CD [36]. An increase in VL is associated with increased absorptive surface area, leading to increased nutrient absorption [5]. Furthermore, an increase in intestinal epithelial cell turnover is associated with increases in the rate of cell proliferation and migration, leading to increased CD [37] and increased gut energy requirement [36]. It is well documented that dietary vitamin D sources have been shown to enhance the small intestine morphology and immunity of broilers subjected to coccidiosis [7,38]. Prior to a coccidiosis challenge, the in ovo administration of 2.4 μg of 25OHD_3_ has been previously shown to increase VCR in all intestinal sections and decease duodenal and ileal CD in comparison to the other treatments that were investigated in the current study [17]. In comparison to the other in ovo treatments, the 25OHD_3_ alone treatment allowed for a more rapid improvement in small intestine morphology in coccidiosis challenged broilers.

In comparison to D_3_ at the same level of activity, dietary 25OHD_3_ increases VL and decreases CD when used in broiler diets [7]. Additionally, the VL of embryos and newly hatched broilers increased in response to the supplementation of maternal diets with 25OHD_3_ [14]. The effects in response to 25OHD_3_ may be due to the increased intestinal expression of 1α-hydroxylase and vitamin D receptors (VDR) [39]. The enzyme 1α-hydroxylase converts 25OHD_3_ to 1,25-dihydroxyvitamin D_3_ [1,25(OH)_2_ D_3_], which is the active form of vitamin D_3_. Increased levels of 1,25(OH)_2_ D_3_ alone, with or without associated changes in VDR, resulted in an increase in breast meat yield [12] and improvements in the intestinal development [40] and immunity [41] of broilers. It is also worth mentioning that the expression of 1α-hydroxylase remains constant during systemic inflammation [42]. Coccidiosis has been shown to cause systemic inflammation in broiler chickens [8]. Additionally, in comparison to diluent- or D_3_-injected treatment groups in a companion study, the expression of 1α-hydroxylase and immune suppressor genes increased 14 d post-challenge in broilers that were in ovo-injected with of 25OHD_3_ (unpublished data). Therefore, improvements in broiler small intestine morphology during a coccidiosis challenge in response to 25OHD_3_ alone might have been due to increased intestinal expression of 1α-hydroxylase and an enhanced immunity. 

In the current study, the in ovo injection of 25OHD_3_ reduced NO production during the coccidiosis challenge. Nitric oxide is an important indicator of on inflammatory response and is produced by the oxidation of L-arginine by the action of NO synthase. Inducible NO synthase is considered more important in immunity and inflammation [43]. The expression of cytokines such as IL-1β, IL-12, interferon-γ, and tumor necrosis factor have been shown to increase the production of NO [44]. In addition, *Eimeria*-infected chickens produced higher serum NO concentrations compared with non-infected controls [45]. Dietary 25OHD_3_ reduced pro-inflammatory responses and increased anti-inflammatory response genes during a coccidiosis infection in comparison to vitamin D_3_ at the same level of activity [13]. In addition, 25OHD_3_ modulates the effects of immunoregulatory T-cells during a coccidiosis infection. Dietary 25OHD_3_ has been shown to increase the number of CD4 + CD25 + cells in mucosal regions, such as cecal tonsils, of *Eimeria*-infected chickens [9]. These results indicate that 25OHD_3_ has the potential to reduce bird losses due to a compromised immunity caused by a coccidia challenge.

## 5. Conclusions

In conclusion, the aim of this study was to determine the small intestine morphology and inflammatory response of broilers challenged with coccidiosis when birds were injected in ovo with two vitamin D_3_ sources. Our findings suggest that coccidiosis has detrimental effects on the small intestine morphology and immunity of broilers. However, the in ovo injection of 2.4 μg of 25OHD_3_ reduced NO production and small intestine CD and increased the small intestine VL and VCR of challenged broilers. However, the other in-ovo-injected treatments tested in this study did not exhibit beneficial effects on the small intestine morphology and inflammatory response of the challenged birds. These results therefore indicate that in ovo injection of 25OHD_3_ may reduce negative effects on the inflammation and small intestine morphology of broilers caused by coccidiosis. These benefits may have been due to the expression of genes that contributed to the improved intestinal development and subsequent health of the broilers. Further research is required to determine the regulatory effects of vitamin D_3_ sources on immunity and intestinal development during a coccidiosis infection. 

## Figures and Tables

**Table 1 animals-12-01027-t001:** Effects of in ovo injection treatment (non-injected, diluent-injected (50 μL) and 50 μL of diluent containing 2.4 μg of vitamin D_3_ (D_3_), 2.4 μg of 25-hydroxycholecalciferol (25OHD_3_), or 2.4 μg of D_3_ and 2.4 μg of 25OHD_3_ (D_3_ + 25OHD_3_) on BW, lengths of small intestine sections, and weights of the liver, spleen, bursa, duodenum, jejunum, ceca, and ileum as percentages of BW at 14 and 28 d of age (doa).

In Ovo Injection Treatment	BW (g)	Duodenum Length (cm)	Jejunum Length (cm)	Ileum Length (cm)	Ceca Length (cm)	Duodenum Weight (%)	Jejunum Weight (%)	Ileum Weight (%)	Ceca Weight (%)	Spleen (%)	Bursa (%)	Liver (%)
						14 doa						
Non-injected ^1^	424	20.3	39.8	37.9	16.5 ^e^	1.28	1.68	1.87	0.49	0.08	0.19	2.47
Diluent ^2^	433	20.0	39.0	37.2	20.5 ^d^	1.00	1.38	1.59	0.68	0.09	0.21	2.38
D_3_ ^3^	399	20.7	38.7	35.9	24.5 ^c^	1.19	1.64	1.76	0.66	0.08	0.22	2.47
25OHD_3_ ^4^	416	19.9	43.9	42.4	28.5 ^b^	1.14	1.86	2.02	0.62	0.10	0.23	2.29
D_3_ + 25OHD_3_ ^5^	414	19.9	38.2	37.7	32.5 ^a^	0.96	1.74	1.77	0.53	0.07	0.19	2.59
Pooled SEM	19.6	0.71	1.85	1.99	0.001	0.101	0.209	0.177	0.078	0.009	0.018	0.37
*p*-value	0.628	0.889	0.109	0.073	0.001	0.071	0.344	0.418	0.196	0.055	0.334	0.164
						28 doa						
Diluent	1455	27.3	62.3	63.1	13.9 ^b^	0.83	1.54	1.53	0.62	0.089	0.203 ^a^	2.41
D_3_	1417	24.7	68.3	57.3	16.1 ^a^	0.79	1.55	1.56	0.67	0.104	0.212 ^a^	2.33
25OHD_3_	1547	28.8	70.0	67.6	15.4 ^a^	0.78	1.57	1.48	0.61	0.110	0.161 ^b^	2.50
D_3_ + 25OHD_3_	1487	27.6	67.3	62.8	14.8 ^ab^	0.77	1.58	1.35	0.62	0.105	0.212 ^a^	2.52
Pooled SEM	72.2	2.51	3.97	5.62	0.663	0.053	0.128	0.140	0.067	0.0096	0.0198	0.139
*p*-value	0.344	0.436	0.270	0.361	0.016	0.657	0.983	0.456	0.865	0.082	0.049	0.538

^a–e^ Treatment means within the same column within effect with no common superscripts are significantly different (*p* < 0.05). ^1^ Eggs that were not injected with any solution and also were not challenged with coccidial vaccine at 14 doa. ^2^ Eggs injected with 50 μL commercial diluent at d 18 of incubation and also were challenged with coccidial vaccine at 14 doa. ^3^ Eggs injected with 50 μL commercial diluent containing vitamin 2.4 μg of D_3_ at d 18 of incubation and also were challenged with coccidial vaccine at 14 doa. ^4^ Eggs injected with 50 μL commercial diluent containing 2.4 μg of 25OHD_3_ at d 18 of incubation and also were challenged with coccidial vaccine at 14 doa. ^5^ Eggs injected with 50 μL commercial diluent containing 2.4 μg of D_3_ and 2.4 μg of 25OHD_3_ at d 18 of incubation and also were challenged with coccidial vaccine at 14 doa.

**Table 2 animals-12-01027-t002:** Effects of in ovo injection treatment (non-injected, diluent-injected (50 μL) and 50 μL of diluent containing 2.4 μg of vitamin D_3_ (D_3_), 2.4 μg of 25-hydroxycholecalciferol (25OHD_3_), or 2.4 μg of D_3_ and 2.4 μg of 25OHD_3_ (D_3_ + 25OHD_3_) on plasma nitric oxide (NO) and IL-1β concentrations.

In Ovo Injection Treatment	NO-d14 ^1^	NO-d28 ^2^	IL-1β-d14 ^3^	IL-1β-d28 ^4^
	μM			
Non-injected ^5^	13.92	-	8.50	-
Diluent ^6^	14.36	18.6 ^a^	8.26	10.39
D_3_ ^7^	14.34	18.5 ^a^	6.65	10.60
25OHD_3_ ^8^	11.08	16.1 ^b^	6.87	6.85
D_3_ + 25OHD_3_ ^9^	15.66	18.1 ^a^	6.23	9.10
Pooled SEM	1.711	0.691	0.826	2.01
*p*-value	0.432	0.028	0.220	0.541

^a,b^ Treatment means within the same column within effect with no common superscripts are significantly different (*p* < 0.05). ^1^ Plasma nitric oxide concentration at 14 d of age (doa). ^2^ Plasma nitric oxide concentrations at 28 doa. ^3^ Plasma IL-1β concentrations at 14 doa. ^4^ Plasma IL-1β concentrations at 28 doa. ^5^ Birds that were not injected with any solution and also were not challenged with coccidial vaccine at 14 doa. ^6^ Eggs injected with 50 μL commercial diluent at d 18 of incubation and also were challenged with coccidial vaccine at 14 d of age. ^7^ Eggs injected with 50 μL commercial diluent containing vitamin 2.4 μg of D_3_ at d 18 of incubation and also were challenged with coccidial vaccine at 14 doa. ^8^ Eggs injected with 50 μL commercial diluent containing 2.4 μg of 25OHD_3_ at d 18 of incubation and also were challenged with coccidial vaccine at 14 doa. ^9^ Eggs injected with 50 μL commercial diluent containing 2.4 μg of D_3_ and 2.4 μg of 25OHD_3_ at d 18 of incubation and also were challenged with coccidial vaccine at 14 doa. NO-d28 and IL-1β-d28 were not measured for the non-injected treatment.

**Table 3 animals-12-01027-t003:** Effects of in ovo injection treatment (non-injected, diluent-injected (50 μL) and 50 μL of diluent containing 2.4 μg of vitamin D_3_ (D_3_), 2.4 μg of 25-hydroxycholecalciferol (25OHD_3_), or 2.4 μg of D_3_ and 2.4 μg of 25OHD_3_ (D_3_ + 25OHD_3_) on small intestine morphology at 14 d of age (doa).

Item			Duodenum				Jejunum						Ileum		
Treatment	Villus Length	Villus Width	Crypt Depth	VCR ^1^	VSA ^2^	Villus Length	Villus Width	Crypt Depth	VCR	VSA	Villus Length	Villus Width	Crypt Depth	VCR	VSA
	μm			Length/Depth	mm^2^	μm			Length/Depth	mm^2^	μm			Length/Depth	mm^2^
Non-injected ^3^	889	81	89	10.1 ^b^	0.29	558	80	86	6.7 ^b^	0.46	171	37	40	4.3 ^c^	0.71
Diluent ^4^	863	96	130	8.1 ^b^	0.37	647	87	88	7.4 ^ab^	0.43	176	40	38	4.8 ^bc^	0.73
D_3_ ^5^	887	84	93	9.7 ^b^	0.30	618	90	101	6.3 ^b^	0.47	193	39	41	4.6 ^bc^	0.70
25OHD_3_ ^6^	1009	93	71	14.7 ^a^	0.29	669	82	78	8.9 ^a^	0.39	237	45	36	4.6 ^a^	0.62
D_3_ + 25OHD_3_ ^7^	945	81	103	9.4 ^b^	0.27	634	90	84	7.6 ^ab^	0.46	191	41	35	6.6 ^b^	0.65
Pooled SEM	50.1	5.2	20.1	0.91	0.027	38.7	4.8	5.3	0.57	0.029	17.4	4.2	2.2	0.36	0.055
*p*-value	0.263	0.146	0.076	0.001	0.102	0.336	0.455	0.056	0.029	0.389	0.087	0.771	0.260	0.001	0.620

^a–c^ Treatment means within the same column within effect with no common superscripts are significantly different (*p* < 0.05). ^1^ Ratio of villus length to crypt depth (VCR). ^2^ Villus surface area (VSA) calculated with average villus length and width = 2π × (width/2) × length. ^3^ Eggs that were not injected with any solution and also were not challenged with coccidial vaccine at 14 doa. ^4^ Eggs injected with 50 μL commercial diluent at d 18 of incubation and also were challenged with coccidial vaccine at 14 doa. ^5^ Eggs injected with 50 μL commercial diluent containing vitamin 2.4 μg of D_3_ at d 18 of incubation and also were challenged with coccidial vaccine at 14 doa. ^6^ Eggs injected with 50 μL commercial diluent containing 2.4 μg of 25OHD_3_ at d 18 of incubation and also were challenged with coccidial vaccine at 14 doa. ^7^ Eggs injected with 50 μL commercial diluent containing 2.4 μg of D_3_ and 2.4 μg of 25OHD_3_ at d 18 of incubation and also were challenged with coccidial vaccine at 14 doa.

**Table 4 animals-12-01027-t004:** Effects of in ovo injection treatment (non-injected, diluent-injected (50 μL) and 50 μL of diluent containing 2.4 μg of vitamin D_3_ (D_3_), 2.4 μg of 25-hydroxycholecalciferol (25OHD_3_), or 2.4 μg of D_3_ and 2.4 μg of 25OHD_3_ (D_3_ + 25OHD_3_) on small intestine morphology at 28 d of age (doa).

Item			Duodenum				Jejunum						Ileum		
Treatment	Villus Length	Villus Width	Crypt Depth	VCR ^1^	VSA ^2^	Villus Length	Villus Width	Crypt Depth	VCR	VSA	Villus Length	Villus Width	Crypt Depth	VCR	VSA
	μm			Length/Depth	mm^2^	μm-			Length/Depth	mm^2^	μm			Length/Depth	mm^2^
Diluent ^3^	918 ^b^	110	103	8.5 ^b^	0.34	734	92	115	6.6	0.42	229	46	39	5.7 ^bc^	0.67
D_3_ ^4^	894 ^b^	95	106	8.5 ^b^	0.33	708	94	106	6.6	0.43	229	42	49	4.6 ^c^	0.63
25OHD_3_ ^5^	1233 ^a^	95	112	12.3 ^a^	0.28	889	109	107	8.5	0.39	313	42	39	8.1 ^a^	0.47
D_3_ + 25OHD_3_ ^6^	983 ^b^	107	97	9.8 ^b^	0.31	729	94	109	6.7	0.43	260	48	38	7.0 ^ab^	0.57
Pooled SEM	54.7	6.9	9.1	0.91	0.030	106.3	10.2	14.4	0.60	0.052	33.0	3.7	4.5	0.743	0.080
*p*-value	0.002	0.524	0.910	0.001	0.268	0.316	0.322	0.930	0.058	0.880	0.056	0.793	0.054	0.001	0.093

^a–c^ Treatment means within the same column within effect with no common superscripts are significantly different (*p* < 0.05). ^1^ Ratio of villus length to crypt depth. ^2^ Villus surface area (VSA) calculated with average villus length and width = 2π × (width/2) × length. ^3^ Eggs injected with 50 μL commercial diluent at d 18 of incubation and also were challenged with coccidial vaccine at 14 doa. ^4^ Eggs injected with 50 μL commercial diluent containing vitamin 2.4 μg of D_3_ at d 18 of incubation and also were challenged with coccidial vaccine at 14 doa. ^5^ Eggs injected with 50 μL commercial diluent containing 2.4 μg of 25OHD_3_ at d 18 of incubation and also were challenged with coccidial vaccine at 14 doa. ^6^ Eggs injected with 50 μL commercial diluent containing 2.4 μg of D_3_ and 2.4 μg of 25OHD_3_ at d 18 of incubation and also were challenged with coccidial vaccine at 14 doa.

## Data Availability

None of the data were deposited in an official repository.

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
