# Peer review of "Effects of the In Ovo Injection of Vitamin D3 and 25-Hydroxyvitamin D3 in Ross 708 Broilers Subsequently Challenged with Coccidiosis: II Immunological and Inflammatory Responses and Small Intestine Histomorphology"

_animals, 2022, doi:10.3390/ani12081027_

Round 1

Reviewer 1 Report

This is generally well written review. I have just a few minor editorial comments.

L1-12 According to the journal's guidelines all information regarding the founding and potential conflicts of interest should be placed in special sections at the end of the manuscript.

L61 in [6] the word “possibly” is used to explain the relation between crypt depth and intestinal energy requirements.

L99-107 It is not easy to follow. The information that 50 eggs per tray were assigned to one of the 5 experimental group (n=10 each per tray) should be clearly given.

L107 I assume that 4 male broilers form each incubator tray level were selected

L130 “lengths and weights of …”

L128-134. Reorganize. One might think that GIT samples were collected first, and then blood was taken.

L136 Please provide the information about ELISA kit manufacturer.

L137-139 I believe that is a standard ELISA procedure. the information that it has been done according to the manufacturer’s protocol is sufficient.

L149 the information what staining was applied to assess the intestinal morphology is needed.

L163 Please changed “observed” to “calculated”

Please correct the Eq. 2 (extra +)

L177 … where Yij represents the dependent variable ..

Please also please define the trend p-values (probably  0.05 < p-value < 0.1)

Table 1 - please add column/row indicating which values represent 14 doa and which 28 doa. P in “P-value” should be written italics.

In methods section, villus length to crypt depth ratio is defined as RVC, in the Results section as VCR.

Author Response

L1-12 According to the journal's guidelines all information regarding the founding and potential conflicts of interest should be placed in special sections at the end of the manuscript.

Answer: Thank you for the suggestion. The relevant information was inserted on Line 411-421 regarding “funding and potential conflicts of interest”

L61 in [6] the word “possibly” is used to explain the relation between crypt depth and intestinal energy requirements.

Answer: Authors are unable to find the word “possibly” in the submitted draft of the manuscript. However, if the suggestion is to add this word in the sentence on line 61, the authors prefer to use the current version because there is strong relationship between an increase in crypt depth and enterocyte turnover, which results in an incremental increase in energy requirements.   

L99-107 It is not easy to follow. The information that 50 eggs per tray were assigned to one of the 5 experimental group (n=10 each per tray) should be clearly given.

Answer: The total number of eggs incubated was 1,000. We had a total of 5 treatments randomly arranged throughout the 20 incubational tray levels. In each incubational tray we had 50 eggs, with a total of 20 incubational trays, Therefore, 50*20=1,000 eggs that were incubated. This information is now presented as “Fifty eggs were assigned to each of 5 pre-assigned treatment groups on that were randomly distributed among 20 incubator tray levels” on lines 99-100

L107 I assume that 4 male broilers form each incubator tray level were selected

Answer: It is at the time of placement in the battery cages. At hatch, all birds for each treatment group were pooled and then 4 male broilers were randomly placed in the battery cages. The relevant information to address this part is inserted on Lines 107-108 “chicks from each replicate basket were pooled within their respective treatment group”

L130 “lengths and weights of …”

Answer: On lines 132-133 the relative information for lengths and weights of the organs that was determined is stated as “lengths and weights of the duodenum, jejunum, ileum and cecum, and weights of the bursa, spleen, liver, and entire small intestine of the each bird were obtained.”

L128-134. Reorganize. One might think that GIT samples were collected first, and then blood was taken.

Answer: Birds were first weighed, and later blood samples were collected. Subsequently, from the same sampled bird, its organs, including some part of the GIT was collected. 

L136 Please provide the information about ELISA kit manufacturer.

Answer: The relevant information was inserted on Line 138

L137-139 I believe that is a standard ELISA procedure. the information that it has been done according to the manufacturer’s protocol is sufficient.

Answer: This assay was specific to measure IL-1β and some specific part of the ELISA assay is specific to IL-1β. For the general part of the assay, the corresponding source that had previously used this procedure was cited.

L149 the information what staining was applied to assess the intestinal morphology is needed.

Answer: Periodic acid-schiff (PAS) and Alcian blue (ALB) stains were used for the SI samples. However, we used the exact similar procedure that was described by Wang et al., 2015. Therefore, we cited this paper for the methodology used in this section

L163 Please changed “observed” to “calculated”

Answer: The relevant correction was applied on Line 162

Please correct the Eq. 2 (extra +)

Answer: The relevant correction was applied on Line 175

L177 … where Yij represents the dependent variable ..

Answer: Yes, it is common to describe the fixed variables (independent variables) of the model because depended variables can be extended to all variables that are measured in the whole study

Please also please define the trend p-values (probably  0.05 < p-value < 0.1)

Answer: As is previously stated in the statistical section “Treatment differences were deemed significant at P≤ 0.05”. We did not consider any treatment differences that had a P-value greater than “0.05”.    

Table 1 - please add column/row indicating which values represent 14 doa and which 28 doa. P in “P-value” should be written italics.

Answer: The relevant correction was applied

In methods section, villus length to crypt depth ratio is defined as RVC, in the Results section as VCR.

Answer: Thank you for the suggestion. The relevant correction was applied on Line 151

Reviewer 2 Report

The authors found that in ovo injection of 2.4 μg of 25OHD3 at 18 day of 24 incubation can enhance intestinal histomorphology as well as inflammatory reaction of broilers when infected with coccidiosis. This research was interesting. However, there some improvement and clarification should be made before acceptance.

1. Line 32. The authors stated that “Only in ovo-injected treatments were challenged with a 20x live coccidial vaccine dosage at 14 d of age (doa).” While in the results part (line 200), we can see that eggs not injected also were challenged with coccidial vaccine. Besides, in the methods part (Line 121), the authors stated that these birds were challenged with live coccidiosis. Coccidiosis or coccidial vaccine was the same in the experiment? Besides, what the meaning of 20 x?

2. Line 31. “At hatch, 4 male chicks were randomly assigned to each of 40 battery cages.” Line 107-108 “4 male broilers were randomly selected and placed in 8 isolated replicate wire-floored battery cages for each treatment in 2 separate rooms (320 total birds).” The description was not clear, it is difficult to understand. Line 128-129. How many birds were selected in each treatment to sampling? The present description was not obvious.

3. Line 131. Why you use the relative weight in the research? Whether you have some references using the same way?

4. Line 208-209. The authors stated that “No significant treatment effects were observed for the immunological measurements at 14 doa (before coccidiosis challenge) or 28 doa (after coccidiosis challenge)”. The description was not accurate. Because there had significant effect on plasma NO concentration at 28 doa.

5. Line 295. What is the meaning of BWG?

6. Line 346. The authors concluded that “Our findings suggest that coccidiosis has detrimental effects on the small intestine morphology and immunity of broilers.” According the description in the results part, all the birds were challenged with coccidiosis. It is difficult to come to the above conclusion.

Author Response

The authors found that in ovo injection of 2.4 μg of 25OHD3 at 18 day of 24 incubation can enhance intestinal histomorphology as well as inflammatory reaction of broilers when infected with coccidiosis. This research was interesting. However, there some improvement and clarification should be made before acceptance.

  1. Line 32. The authors stated that “Only in ovo-injected treatments were challenged with a 20x live coccidial vaccine dosage at 14 d of age (doa).” While in the results part (line 200), we can see that eggs not injected also were challenged with coccidial vaccine. Besides, in the methods part (Line 121), the authors stated that these birds were challenged with live coccidiosis. Coccidiosis or coccidial vaccine was the same in the experiment? Besides, what the meaning of 20 x?

Answer: Thank you for the comments. In the current draft on Lines 125-126, it was mentioned that we did not challenge the birds that belonged to the non-injected treatment group. Also, in the results section, we used the “unchallenged” term for the non-injected treatment group, and in the table this group is now defined as “Birds that were not injected with any solution and that were also not challenged with the coccidial vaccine at 14 doa”.

To address the second part of question, 20X means 20 time dose of vaccine in order to cause the coccidiosis challenge. A 1x dosage is recommended for vaccination against disease and in this particular research is coccidiosis. In consideration of the fact that considering there is an optimum level of vaccine to stimulate non-lethal disease, the chance of occurrence of disease is more likely when a greater dose of the vaccine is used.

  1. Line 31. “At hatch, 4 male chicks were randomly assigned to each of 40 battery cages.” Line 107-108 “4 male broilers were randomly selected and placed in 8 isolated replicate wire-floored battery cages for each treatment in 2 separate rooms (320 total birds).” The description was not clear, it is difficult to understand. Line 128-129. How many birds were selected in each treatment to sampling? The present description was not obvious.

Answer: a) At placement, we had an overall number of 320 birds in 2 rooms and in each room there were 4 birds in each cage that were randomly allocated to 5 in ovo injected treatments, resulting in 8 replications per treatment. the relevant information was inserted on lines 108-111 b) To address the second part, on Line 130-131, it is now mentioned a that total of 40 birds were used in each treatment-replicate cage. The relevant information was inserted on line 130.

  1. Line 131. Why you use the relative weight in the research? Whether you have some references using the same way?

Answer: It is a very common practice to report organ weights relative to body weight. A bigger bird may have bigger organs, thus, presentation of the proportion of organ weight to body weight of the birds will correct for body weight as a confounding facor. Therefore, the use of relative weight is always recommend   

  1. Line 208-209. The authors stated that “No significant treatment effects were observed for the immunological measurements at 14 doa (before coccidiosis challenge) or 28 doa (after coccidiosis challenge)”. The description was not accurate. Because there had significant effect on plasma NO concentration at 28 doa.

Answer: Thank you for the suggestion. The relevant correction was performed on Line 210.

  1. Line 295. What is the meaning of BWG?

Answer: BWG is body weight gain, and BWG is a standard term. 

  1. Line 346. The authors concluded that “Our findings suggest that coccidiosis has detrimental effects on the small intestine morphology and immunity of broilers.” According the description in the results part, all the birds were challenged with coccidiosis. It is difficult to come to the above conclusion.

Answer: As was addressed before, in the current draft on lines 125-126, it was mentioned that we did not challenge the birds that belonged to the non-injected treatment group. Also, in the results section, we used “unchallenged” as a term for the non-injected treatment group, and in the table this group is now defined as “Birds that were not injected with any solution and that were also not challenged with the coccidial vaccine at 14 doa”.